# NephroPOC - Risk assessment and prediction of acute kidney injury in emergency patients with suspected organ dysfunction: Secondary analysis from the prospective observational LifePOC study

Caroline Neumann[1]*, Margit Leitner[1], Thomas Lehmann[2], Michael Kiehntopf[3], Michael Joannidis[4], Myrto Bolanaki[5], Anna Slagman[5], Martin Möckel[5], Michael Bauer[1], Johannes Winning[1,6]

**1** Department of Anaesthesiology and Intensive Care Medicine, University Hospital Jena, Am Klinikum 1, Jena, Germany, **2** Centre for Clinical Studies, Jena University Hospital, Am Klinikum 1, Jena, Germany, **3** Institute of Clinical Chemistry and Laboratory Diagnostics and Integrated Biobank Jena (IBBJ), Jena University Hospital, Am Klinikum 1, Jena, Germany, **4** Division of Intensive Care and Emergency Medicine, Department of Internal Medicine, Medical University Innsbruck, Anichstrasse 35, Innsbruck, Austria, **5** Charité – Universitätsmedizin Berlin, corporate member of Freie Universität Berlin and Humboldt Universität zu Berlin, Department of Emergency and Acute Medicine, Augustenburger Platz 1, Berlin, Germany, **6** University of Applied Sciences Jena, Carl-Zeiss-Promenade 2, Jena, Germany

* caroline.neumann@med.uni-jena.de

## Abstract

We aimed to assess markers and risk factors for imminent acute kidney injury (AKI) in emergency patients, as risk stratification in the emergency department is currently not widely used. Using data from a sub-cohort (440 patients) of the prospective multicentre LifePOC study (1434 patients), proenkephalin A 119−159 (penKid) was assessed for early identification of subclinical kidney damage compared to serum creatinine in emergency patients with a qSOFA score ≥1. Logistic regression was applied to assess the usefulness of penKid, four further biomarkers (midregional pro-adrenomedullin, bioactive adrenomedullin, dipeptidyl-peptidase-3, procalcitonin) and clinical risk factors to predict AKI within 24 h, 48 h and 72 h after admission, need for organ support and 28-day mortality. PenKid and bio-adrenomedullin performed moderately to predict AKI within 48 h (AUC 0.645, 95% CI: 0.582–0.703 and AUC 0.647, 95% CI: 0.583–0.707, respectively). Pre-existing chronic kidney disease (OR 2.36, 95% CI: 1.06–5.27), confirmed sepsis (OR 2.41, 95% CI: 1.28–4.56), mechanical ventilation (OR 3.03, 95% CI: 1.48–6.19), and elevated levels of penKid (OR 2.21, 95% CI: 1.60–3.07) at admission were associated with an increased risk of AKI whereas a restrictive fluid management (OR 0.43, 95% CI: 0.26–0.71) was associated with a lower risk of AKI. Patients at high AKI risk may be identified based on specific risk factors, bio-ADM and penKid. The trial was registered in the German Registry for Clinical Trials (DRKS00011188) on 20 October 2016.

**Data availability statement:** The data used for the presented analysis contains sensitive patient information, requiring restricted access in line with legal restrictions and patient consent. Data can be made available upon reasonable request to the Department of Anesthesiology and Intensive Care Medicine, Jena University Hospital (e-mail: KAI-Chefsekretariat@med.uni-jena.de).

**Funding:** This study was funded by the Federal Ministry of Education and Research (Grant No. 03ZZ0810B) within the INFECTCONTROL framework. Project beneficiaries are AS, MM, MB, JW, MB.

**Competing interests:** The authors have declared that no competing interests exist.

## Introduction

Acute kidney injury (AKI) summarises heterogeneous conditions, which are characterized by a sudden decrease in glomerular filtration rate (GFR). Signs of this type of kidney injury, which occurs in approximately 10–20% of hospitalized patients, are increased serum creatinine (Scr) concentration and/or oliguria [1,2]. Poor management of the syndrome can in part be attributed to a lack of awareness among clinicians regarding early recognition and an absence of standards for prevention and intervention [3]. Early diagnosis and identification of the underlying aetiology, e.g., infection/sepsis, are essential to guide management. Basing AKI diagnosis on an increase in serum creatinine (Scr) and/or a reduction of urine output (UO) suffers from several shortcomings [4].

Current evidence from clinical studies supports the use of new biomarkers in the detection, prevention, and management of AKI in different patient cohorts, such as damage biomarkers [5–9], stress biomarkers [10,11], and functional markers, e.g., Scr, Cystatin C [6] and proenkephalin A 119−159 (penKid) [12–14]. In this study, we investigated the performance of penKid for early identification of ED patients at risk to develop AKI as compared to Scr. We included further biomarkers, indicative of different underlying pathologies, namely adrenomedullin (ADM), dipeptidylpeptidase-3 (DPP-3) and procalcitonin (PCT) as well as clinical factors to assess the predictive value of these markers or their combinations for AKI in patients with differing disease severities. The glomerular filtered penKid is a stable prohormone fragment of the enkephalin family, with a long in vivo half-life, which is not influenced by sex or age. Studies have shown that penKid strongly correlates with kidney function and the measured GFR [15,16]. ADM, either as the midregional fragment of the prohormone (MR-proADM) or as its bioactive form (bio-ADM), is a hormone that regulates endothelial function, relevant for keeping vascular integrity [17,18]. ADM is also produced in the kidneys, in particular under hypoxic conditions [19]. Elevated levels of ADM therefore might be an early sign of renal damage. DPP-3 is an enzyme that catalyses the breakdown of various peptide hormones, e.g., angiotensin II and enkephalines. It has wide-ranging biological functions in blood pressure regulation and the pro-inflammatory response and acts as a regulator of the cellular oxidative stress response pathway [20]. The enzyme also is a key modulator of the renin-angiotensin system and might thus be a valuable biomarker in cardiovascular and renal pathologies. PCT is a precursor to the hormone calcitonin. The usefulness of PCT to terminate anti-infective therapy is supported by guideline recommendations [21]. Studies have confirmed PCT to be a strong marker of inflammation, not only in acute bacterial sepsis but also in other inflammatory disorders. Moreover, PCT predicts the development of AKI in critically ill patients [22], in patients with pancreatitis [23] and contrast-induced AKI [24].

We placed particular emphasis on penKid, given its documented use in the detection of renal dysfunction, as, e.g., in a recently developed formula for GFR estimation [25]. To establish its predictive value compared to other markers, we evaluated the four further biomarkers along with clinical risk factors as predictors of AKI in a cohort

of ED patients presenting with suspected organ dysfunction. Finally, we assessed all biomarkers for their ability to predict secondary endpoints, i.e., need for vasopressors, mechanical ventilation and renal replacement therapy (RRT), hospital and ICU mortality as well as hospital and ICU length of stay (LOS).

## Materials and methods

### Study design and patient characteristics

The NephroPOC study used data from the LifePOC study [26], "German Clinical Trials Register" ID DRKS00011188, an observational multicentre study conducted in three German university hospitals (Jena University Hospital, Charité – University Hospital Berlin, Virchow- Klinikum and Campus Charité Mitte) evaluating 1434 of the 1477 enrolled adult patients (≥18 years) admitted to the emergency department (ED) with suspected organ dysfunction based on a qSOFA score ≥1. Patients were recruited from December 19, 2016, to June 7, 2019. Exclusion criteria were pregnancy, patients with acute myocardial infarction, stroke, a leading trauma diagnosis, or patients with a life expectancy of less than 28 days on admission (diagnosis of an advanced tumour disease or pre-existing or established status of "do not escalate, do not resuscitate"). For this analysis, another exclusion criterion was already existing RRT on admission. LifePOC aimed to establish early markers for imminent sepsis. The study was approved by the institutional review boards of the respective universities. Written informed consent was obtained from all patients or their legal representatives. The study protocol was accepted by local ethical committees and conducted in accordance with directives as well as good clinical practice and Declaration of Helsinki.

### Measurement of biomarkers, serum creatinine and estimation of GFR

After study inclusion, blood samples were collected within 12 hours after initial ED presentation and processed and stored as described previously [26]. Methods for biomarker measurements, physiological ranges and clinical cut-offs as well as GFR estimation are detailed in **Appendix A in S1 Text**.

### Data collection

All relevant data on demographics, comorbidities (recorded in the Charlson Comorbidity Index (CCI) [27]), laboratory findings and physiological variables were extracted, reviewed, and recorded from the hospital database for three calendar days starting from the time of ED presentation. Mortality was assessed up to 28 days. In case of earlier hospital discharge, patients received a follow-up call from study personnel as agreed on enrolment.

### AKI assessment

The primary endpoint was the development of AKI within 48 h. AKI was defined according to the current KDIGO definition by an increase in Scr by 0.3 mg/dl (26.5 µmol/L) or more within 48 hours or an increase in Scr to 1.5 times or higher than the baseline, which is known or presumed to have occurred within the prior 7 days. Prioritizing 48 h AKI as primary outcome aligns with the KDIGO absolute-increase criterion while AKI within 24 and 72 h was treated as supportive secondary windows.

AKI severity staging was also done according the KDIGO definition [28]. Urine output could not be used as an AKI criterion due to an insufficient recording in this cohort all-comers to the ED, mainly of moderate disease severity. Scr was measured in blood samples directly after admission in ED and, although only in a subgroup of patients, 24, 48 and 72 hours after admission. For the primary endpoint, a sub-cohort of n = 440 patients was available with complete Scr data within this timeframe (see **Fig 1**). To illustrate Scr changes over time, we evaluated AKI within 24 and 72 as secondary endpoints for patients with complete Scr data from all days within the respective time window. To address the potential selection bias, we compared the baseline characteristics of the entire cohort with the sub-cohort of patients used for prediction of AKI within 48 h (**Table A in S1 Text**).

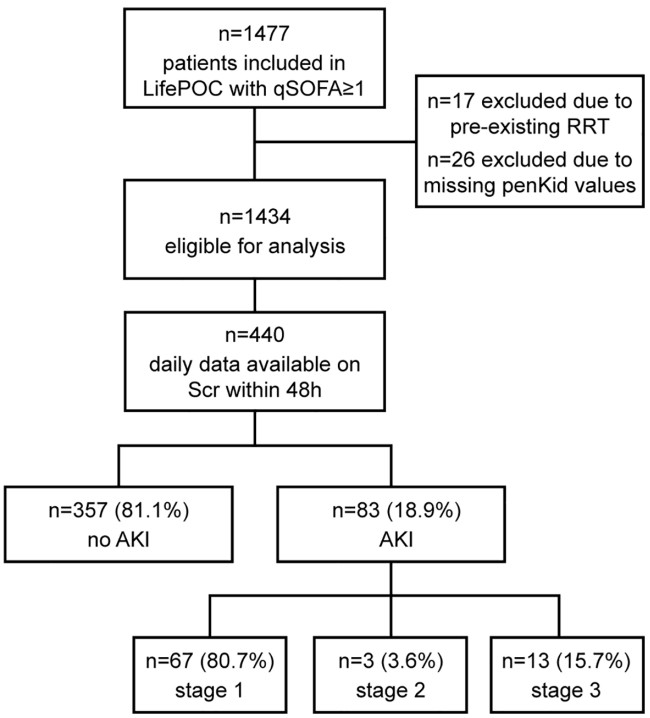

**Fig 1. Patient flow diagram of the whole cohort with and without AKI within 48 hours.** penKid: proenkephalin A 119-159; RRT: renal replacement therapy; Scr: serum creatinine; AKI: acute kidney injury.

### Determination of pre-existing renal impairment and secondary endpoints

The baseline Scr of all patients was the level determined from blood samples taken within 12 hours after ED admission. Normal serum reference values for Scr were regarded as 0.72–1.26 mg/dl for men ≥ 18 years and as 0.57–1.12 mg/dl for women ≥18 years. Values above this range were considered as increased Scr levels. To distinguish between long-standing and acute impairment of kidney function, we used the measured CCI, which was recorded at admission, and in which moderate to severe CKD is included and defined as Scr values above 3 mg/dL (264 µmol/L).

Secondary endpoints were need for vasopressors/inotropes, RRT and mechanical ventilation within 48 hours after admission. Furthermore, hospital and 28-day all-cause mortality and length of stay (LOS) on ICU and in hospital were evaluated.

### Statistical analysis

Values are expressed as medians and interquartile ranges (IQR), or counts and percentages, as appropriate. Group comparisons of continuous variables were performed using the Mann-Whitney U test. Categorical data were compared between groups using Fisher's exact test. Logistic regression was used to evaluate each biomarker's ability to predict AKI within 24 h, 48 h and 72 h. The area under the curve (AUC) is given as an effect measure for uni- and multivariable models.

Net reclassification improvement (NRI) was calculated for comparing eGFR estimates based on CKD-EPI (without pen-Kid) and PENK-Crea (with penKid) based on eGFR categories (cut points 30, 60 and 90) [29].

For a considerable proportion of patients Scr data during their inpatient stay were unavailable, resulting in n = 994 with an unknown status of AKI within 48 h. To address this problem, multiple imputation using the fully conditional specification

(FCS) approach was performed, including all potential predictors and AKI within 48h in the imputation model. Multiple binary logistic regression analysis with backward selection was applied to the imputed datasets to identify the predictors associated with an increased risk of AKI within 48h. Wald statistic was used for stepwise removal of variables from the initial model including all predictors. Odds ratios with 95% confidence intervals are provided for the remaining predictors in the model.

Receiver Operating Characteristic (ROC) analysis was performed on the biomarkers to predict the AKI within 48 hours. The AUC was calculated for each biomarker with 95% confidence intervals. AUC was compared pairwise by the DeLong test; a significance level was set at 0.005 for each test to account for multiple comparisons (Bonferroni correction).

"Least absolute shrinkage and selection operator" (LASSO) regression [30] for variable selection was used to find the combination of biomarkers achieving the best discrimination of patients with and without AKI. The LASSO regression model with logit link function was fitted for the AKI within 48 h including all biomarkers as independent variables. Logarithmic values of the biomarkers were used since the data were not normally distributed. Ten-fold cross validation was applied to optimize the shrinkage parameter λ in the LASSO regression. AUCs with 95% confidence intervals are reported for the model after cross-validation. The same analyses (deLong Test, LASSO regression with all biomarkers) were performed for all binary secondary endpoints. A linear LASSO regression model was applied for the continuous secondary endpoints (ICU and hospital LOS), with the coefficient of determination reported to assess the model fit.

All statistical analysis were performed by R 4.2.2 (R Foundation for Statistical Computing, Vienna, Austria), Statistical Package for the Social Sciences (SPSS) version 22.0 (SPSS Inc., Chicago, Illinois, USA) and with SAS 9.4 (SAS Institute Inc., Cary, 219 NC). All statistical tests were two-sided. A p-value of < 0.05 was considered as statistically significant.

## Results

### Patient characteristics

1477 patients with qSOFA≥1 were included in the LifePOC study, 17 of which were excluded from this post-hoc analysis due to pre-existing RRT and further 26 because of missing penKid values. From these 1434 patients eligible for analysis, serial data on Scr levels within 48 hours and biomarker levels at admission were available in 440 patients. 83 of the 440 patients (18.9%) developed AKI within 48 hours. Of these patients, 67 (80.7%) developed stage 1 AKI, 3 patients stage 2 AKI (3.6%) and 13 (15.7%) patients progressed to stage 3 AKI (**Fig 1**).

Epidemiological and clinical data of the patients, for whom the primary endpoint, AKI within 48 h (n = 440), could be determined, is shown in **Table 1**.

For continuous variables median [25th-75th percentile] and for categorical variables absolute and (relative) frequencies are reported.

The average age did not differ between patients with or without AKI after 48 h and both groups were characterized by SOFA scores consistent with mild disease severity. An infection was confirmed in 72% of the whole cohort within 48 h but did not differ between the groups. In the whole cohort, 51.8% of the patients developed sepsis within 48 hours. Patients with sepsis suffered from AKI within 48 hours more frequently (63.9%) than patients without sepsis (49.0%, p = 0.019), as was the case for septic shock (18.1% in AKI vs. 7.0% non-AKI, p = 0.004). Significantly more patients with AKI required organ support within 48 h (RRT, mechanical ventilation, vasopressors/inotropes; for details see **Table B in S1 Text**). Among the biomarkers measured on admission, levels differed significantly between AKI and non-AKI patients for penKid (107.61 [66.9–212.19] pmol/L vs. 73.51 [46.91–116.76] pmol/L, respectively, p<0.001), MR-proADM (2.23 [1.53–4.18] vs. 1.73 [1.03–3.02], p<0.001) and bio-ADM (89.97 [50.01–140.8] vs. 55.89 [34.44–116.37], p<0.001). eGFR was lower in AKI patients (**Table C in S1 Text**). PenKid levels at admission differed depending on the severity of AKI within 48 h. PenKid

**Table 1. Baseline characteristics of patients at admission with and without AKI within 48 hours and main outcomes within 48 h (n = 440).**

| Patient characteristics | n | Total cohort (n=440, 100%) | Patients with AKI (n= 83, 18.9%) | Patients without AKI (n=357, 81.1%) | p-value |
|---|---|---|---|---|---|
| Age (years) | 440 | 73 [63-79] | 75 [66-81] | 72 [62-79] | 0.112 |
| Males, no. (%) | 440 | 251 (57%) | 47 (56.6%) | 204 (57.1%) | 1.000 |
| **Severity scores** | 439 | | | | |
| qSOFA | 439 | | | | 0.058 |
| 1 | | 333 (75.7%) | 56 (67.5%) | 277 (77.8%) | |
| 2 | | 102 (23.2%) | 27 (32.5%) | 75 (21.1%) | |
| 3 points | | 4 (0.9%) | 0 (0%) | 4 (1.1%) | |
| Charlson Comorbidity Index | 406 | 2 [1-3] | 2 [1-4] | 2 [1-3] | 0.081 |
| CKD | 406 | 40 (9.9%) | 20 (24.1%) | 20 (5.6%) | <0.001 |
| SOFA | 269 | 3 [2-5] | 4 [2-5] | 3 [2-5] | 0.060 |
| Renal SOFA | 440 | 1 [0-1] | 1 [0-2] | 0 [0-1] | **0.003** |
| Body temperature (°C) | 362 | 37.4 [36.8-38.3] | 37.4 [36.8-38.3] | 37.4 [36.7-38.3] | 0.548 |
| Heart rate (bpm) | 433 | 95 [81-110] | 96 [82-110] | 93 [76-109] | 0.231 |
| MAP (mmHg) | 432 | 82.5 [68.75-100] | 82 [69-101] | 88 [66.5-100] | 0.879 |
| RR (per minute) | 437 | 25 [22-28] | 25 [23-28] | 25 [22-28] | 0.637 |
| paO2/FiO2 ratio | 412 | 333 [267-410] | 309 [231-348] | 348 [271-447] | **0.001** |
| GCS | 437 | 15 [15-15] | 15 [15-15] | 15 [15-15] | 0.707 |
| **Laboratory variables and biomarkers** | | | | | |
| pH | 395 | 7.4 [7.35-7.45] | 7.4 [7.35-7.44] | 7.4 [7.34-7.45] | 0.763 |
| SBE (mmol/L) | 396 | -0.1 [-3.12-2.52] | -0.2 [-3.25-2.4] | 0 [-2.3-2.7] | 0.266 |
| Bilirubin (µmol/L) | 299 | 0.67 [0.41-1.12] | 0.69 [0.41-1.12] | 0.65 [0.44-1.06] | 0.631 |
| Lactate (mmol/L) | 403 | 1.9 [1.32-2.7] | 1.9 [1.32-2.72] | 1.83 [1.32-2.5] | 0.604 |
| CRP (mg/L) | 438 | 70.9 [18.45-177.6] | 67.6 [18.25-179.45] | 88.9 [20.65-152.1] | 0.842 |
| Leukocytes (Gpt/L) | 439 | 11.7 [8.55-15.68] | 11.95 [8.57-15.78] | 11 [8.55-14.73] | 0.380 |
| Thrombocytes (Gpt/L) | 439 | 229 [167-307.5] | 235 [170-310] | 212 [151-293] | 0.088 |
| **Biomarkers** | | | | | |
| Scr (mg/dL) | 440 | 1.24 [0.88-1.85] | 1.49 [1.13-2.25] | 1.17 [0.87-1.83] | **0.007** |
| penKid (pmol/L) | 440 | 79.38 [49.4-126.76] | 107.61 [66.9-212.19] | 73.51 [46.91-116.76] | **<0.001** |
| MR-proADM (nmol/L) | 440 | 1.82 [1.11-3.28] | 2.23 [1.53-4.18] | 1.73 [1.03-3.02] | **<0.001** |
| bio-ADM (pg/mL) | 440 | 63.37 [36.7-121.52] | 89.97 [50.01-140.8] | 55.89 [34.44-116.37] | **<0.001** |
| DPP-3 (ng/mL) | 440 | 18.69 [12.51-32.66] | 19.12 [12.71-33.7] | 18.68 [12.5-31.74] | 0.539 |
| PCT (µg/L) | 440 | 0.32 [0.11-2.2] | 0.49 [0.15-4.14] | 0.27 [0.11-1.78] | 0.051 |
| **Admission diagnosis** | 440 | | | | |
| Pulmonary diseases | | 169 (38.4%) | 33 (39.8%) | 136 (38.1%) | 0.803 |
| Cardiovascular diseases | | 112 (25.5%) | 20 (24.1%) | 92 (25.8%) | 0.889 |
| Sepsis | | 76 (17.3%) | 18 (21.7%) | 58 (16.2%) | 0.259 |
| Diseases of UGT | | 95 (21.6%) | 19 (22.9%) | 76 (21.3%) | 0.768 |
| Diseases of digestive system | | 41 (9.3%) | 2 (2.4%) | 39 (10.9%) | **0.012** |
| Solid tumours | | 11 (2.5%) | 2 (2.4%) | 9 (2.5%) | 1.000 |
| Others | | 118 (26.8%) | 83 (18.9%) | 357 (81.8%) | 0.584 |
| **Focus of infection within 48 h** | 440 | | | | |
| Pulmonary | | 174 (39.5%) | 29 (34.9%) | 145 (40.6%) | 0.384 |
| Intraabdominal | | 46 (10.5%) | 5 (6.0%) | 41 (11.5%) | 0.167 |
| Urogenital | | 58 (13.2%) | 16 (19.3%) | 42 (11.8%) | 0.074 |

*(Continued)*

**Table 1.** (Continued)

| Patient characteristics | n | Total cohort (n=440, 100%) | Patients with AKI (n= 83, 18.9%) | Patients without AKI (n=357, 81.1%) | p-value |
|---|---|---|---|---|---|
| Cardiovascular | | 5 (1.1%) | 1 (1.2%) | 4 (1.1%) | 1.000 |
| Cerebral | | 3 (0.7%) | 1 (1.2%) | 2 (0.6%) | 0.467 |
| Skin and soft tissue | | 20 (4.5%) | 4 (4.8%) | 16 (4.5%) | 0.778 |
| Primary bacteraemia | | 4 (0.9%) | 2 (2.4%) | 2 (0.6%) | 0.163 |
| Others | | 7 (1.6%) | 2 (2.4%) | 5 (1.4%) | 0.621 |
| Unknown | | 25 (5.7%) | 5 (6.0%) | 20 (5.6%) | |
| None | | 98 (22.3%) | 18 (21.7%) | 80 (22.4%) | 1.000 |
| **Secondary outcomes within 48 h** | 440 | | | | |
| Sepsis | 433 | 228 (51.8%) | 53 (63.9%) | 175 (49.0%) | **0.019** |
| Septic shock | 432 | 40 (9.1%) | 15 (18.1%) | 25 (7.0%) | **0.004** |

**Abbreviations:** qSOFA: quick Sequential Organ Failure Assessment; CKD: chronic kidney disease; SOFA: Sequential Organ Failure Assessment; MAP: mean arterial pressure; RR: respiratory rate; GCS: Glasgow Coma Scale; SBE: standard base excess; Scr: serum creatinine; CRP: C-reactive protein; penKid: proenkephalin A 119−159; MR-proADM: midregional proadrenomedullin; bio-ADM: bioactive adrenomedullin; DPP-3: dipeptidylpeptidase-3; PCT: procalcitonin; UGT: urogenital tract.

levels of patients with AKI stage 3 were significantly higher compared to no AKI and stage 1 AKI (post hoc $p < 0.05$, **Figure A in S1 Text**).

Due to a large number of missing Scr data during inpatient stay, AKI at 48 h could only be defined in 29.8% of patients of the original LifePOC study (**Fig 1**). Missing Scr during follow-up can be assumed not to be random, but rather due to patient discharge, no further Scr measurements due to the patient's good clinical condition and, less often, due to early death. Discrepancies between the groups with and without complete Scr data is illustrated by marked differences in their baseline characteristics regarding comorbidities, organ dysfunction, vital and laboratory parameters and organ support (**Table A in S1 Text**). The 440 patients with known AKI status and sequential Scr data displayed significantly higher disease severity than the patients of the entire cohort of 1434 patients, e.g., documented in their SOFA scores. Regarding sepsis as an important cause of AKI, 17.3% of patients with known AKI status were diagnosed with sepsis at admission in contrast to 6.3% of the patients excluded from further analysis. After 48 h, more than half of the patient in the analysed cohort (52.5%) had developed sepsis, which is in stark contrast to only 17.8% in the group of excluded patients. Higher dependency on organ support within 48 hours, e.g., need for vasopressors/inotropes (18% vs 1.9%, $p < 0.001$), mechanical ventilation (16.4% vs 4.9%, $p < 0.001$) and need for RRT (3.6% vs 0.2%, $p < 0.001$) further indicate an inevitable selection bias (**Table A in S1 Text**).

### Risk factors of AKI

To evaluate possible risk factors for the development of AKI, we selected variables which, from a clinical perspective and based on existing literature [31], were of interest to be included in the multivariate analysis: age, gender, comorbidities (as recorded in CCI), sepsis within 48 h, mechanical ventilation, fluid therapy and antimicrobial therapy on admission. While all five biomarkers were included in the multiple regression of the AKI risk factors, only elevated levels of penKid at admission proved to be a relevant predictor (OR 2.21, 95% CI: 1.60–3.07). Clinical risk factors for AKI were mechanical ventilation (OR 3.03, 95% CI: 1.48–6.19), CKD (OR 2.36, 95% CI: 1.06–5.27) and confirmed sepsis within 48 hours (OR 2.41, 95% CI: 1.28–4.56), whereas restrictive fluid therapy on admission was associated with lower OR of AKI (OR 0.43, 95% CI: 0.26–0.71). A low paO2/FiO2 ratio (OR 0.99; 95% CI: 0.99–1.00) also was identified as a possible risk factor for AKI, with a significant but minor effect (**Table 2**).

**Table 2. Risk factors for the development of AKI, as determined by multivariate logistic regression with backward selection after multiple imputation.**

| Predictor | OR | lower 95% CI | higher 95% CI | p-value |
|---|---|---|---|---|
| Fluid therapy | 0.43 | 0.26 | 0.71 | **0.001** |
| Mechanical ventilation | 3.03 | 1.48 | 6.19 | **0.004** |
| Heart rate | 0.99 | 0.98 | 1.00 | 0.166 |
| paO2/FiO2 | 0.99 | 0.99 | 1.00 | **0.006** |
| MAP | 1.01 | 0.99 | 1.012 | 0.183 |
| Pre-existing CKD | 2.36 | 1.06 | 5.27 | **0.042** |
| DM with end-organ damage | 1.71 | 0.71 | 4.12 | 0.263 |
| PCT | 0.97 | 0.94 | 1.01 | 0.116 |
| Sepsis within 48 h | 2.41 | 1.28 | 4.56 | **0.020** |
| penKid | 2.21 | 1.60 | 3.07 | **<0.001** |

**Abbreviations:** AKI: acute kidney injury; OR: odds ratio; CI: confidence interval; MAP: mean arterial pressure; CKD: chronic kidney disease; DM: diabetes mellitus; PCT: procalcitonin, penKid: proenkephalin A119-159

## Performance of biomarkers as predictors of AKI

We assessed five biomarkers for their potential to predict AKI development compared to the routinely used Scr. Particular emphasis was placed on penKid, given its documented use in the detection of renal dysfunction. In addition, we considered the impact of pre-existing renal dysfunction and disease severity on biomarker performance.

## Comparison of penKid, Scr and eGFR formulas

The well-described indicators for renal dysfunction penKid, Scr and eGFR formulas were compared for their predictive value of AKI development over an observation period of 72 h (**Tables C** and **D in S1 Text**). Prediction of AKI within 48 h was moderate for penKid (AUROC 0.662 [95% CI: 0.599–0.724]), followed by admission Scr (AUROC 0.596 [95% CI: 0.529–0.663]). Both markers showed a lower AUROC for prediction of AKI within 48 h and 72 h compared to the short-term prediction of AKI within 24 h. The novel formula eGFR$_{PENK-Crea}$ performed slightly superior than well-established eGFR$_{CKD-EPI}$ (**Tables C and D in S1 Text**).

eGFR formulas were further compared using net reclassification improvement (NRI) analysis, showing that using the eGFR$_{PENK-Crea}$ equation leads to a better classification in over 10% of all patients (**Table E in S1 Text**). To analyse the agreement between eGFR categories by PENK-Crea and CKD-EPI, we performed Cohen's Kappa analysis. The resulting Kappa coefficients of 0.672 (95% confidence interval: 0.534 to 0.786) in patients with AKI within 48h and 0.626 (95% confidence interval: 0.559 to 0.686) in patients without AKI within 48h indicates substantial agreement between the two classification methods according to commonly accepted interpretation criteria [32].

## Biomarker performance depending on time to AKI onset and disease severity

For the analysis of development of AKI over time, we investigated all biomarkers regarding different times until AKI onset (24, 48, and 72 h). Scr data were available from 755 patients after 24 h, 440 patients after 48 h and 328 patients after 72 h. 53 out of the 755 patients (7.0%) suffered from AKI within 24 h and 104 patients out of the 328 within 72 h (31.7%).

Prognostic performance of all biomarkers decreased considerably over the observation period. penKid resulted in best AUC after 72 h (**Table 3** and **Figure B in S1 Text**), indicating its potential as earlier warning sign of AKI compared to the other biomarkers tested.

**Table 3. AUC of biomarkers to predict AKI within 24, 48, and 72 hours in the whole cohort.**

| Biomarker | All patients – 24 h (n = 755) AUC (95% CI) | All patients – 48 h (n = 440) AUC (95% CI) | All patients – 72 h (n = 328) AUC (95% CI) |
|---|---|---|---|
| penKid | 0.726 (0.650-0.791) | 0.645 (0.582-0.703) | 0.617 (0.555-0.676) |
| MR-proADM | 0.738 (0.650-0.810) | 0.627 (0.549-0.699) | 0.544 (0.464-0.621) |
| Bio-ADM | 0.704 (0.619-0.778) | 0.647 (0.583-0.707) | 0.560 (0.486-0.631) |
| PCT | 0.677 (0.588-0.754) | 0.573 (0.501-0.642) | 0.483 (0.427-0.540) |
| DPP-3 | 0.583 (0.512-0.652) | 0.535 (0.473-0.597) | 0.475 (0.418-0.533) |
| Scr | 0.680 (0.587-0.761) | 0.592 (0.512-0.667) | 0.537 (0.463-0.610) |

**Abbreviations:** AUC: area under the curve; penKid: proenkephalin A119-159; MR-proADM: midregional proadrenomedullin; bio-ADM: bioactive adrenomedullin; PCT: procalcitonin; DPP-3: dipeptidylpeptidase-3; Scr: serum creatinine

To assess whether the severity of the disease influences the predictive ability of the different biomarkers, we examined the entire cohort (n = 440), and the subgroups of patients transferred to the normal ward (n = 240) or to ICU within 48 h (n = 200). Again, the predictive value of all investigated biomarkers for AKI within 48 h was moderate. ROC analysis of the entire cohort (n = 440) resulted in the best AUC for achieving the primary endpoint for penKid (AUC 0.645, 95% CI 0.582–0.703) and bio-ADM (AUC 0.647, 95% CI 0.583–0.707). Discrimination performed significantly better than in the model with PCT (AUC 0.573 [0.501–0.642]), DeLong test: p < 0.005) and DPP3 (AUC 0.535 [0.473–0.597], p < 0.005). No marked differences were observed in the predictive performance of penKid, bio-ADM, pro-ADM and Scr or in the separate consideration of patients on normal ward and ICU (**Fig C** and **Table F in S1 Text**).

In the subcohort stratified by Scr values and known renal dysfunction status at admission (moderate to severe CKD) (**Table G** and **Fig D in S1 Text** for patients with increased Scr levels with CKD), we could confirm the moderate ability of penKid in predicting the primary endpoint, which was superior for patients with Scr values within the normal gender-specific range at admission compared to patients with an increased Scr level without CKD. The best predictive value of penKid was shown for patients with increased Scr levels and a pre-existing moderate to severe CKD, but data and biomarker comparisons are inconclusive in this small group of patients (n = 39). The AUCs for AKI prediction of the different biomarkers dependent on pre-existing renal dysfunction is summarised in **Table G in S1 Text**.

## Performance of biomarkers to predict secondary endpoints

LASSO regression was applied to identify predictive biomarkers for the secondary endpoints RRT, mechanical ventilation and vasopressor/inotropes requirement within 48 hours of admission (**Table 4**). MR pro-ADM displayed the best predictive ability regarding vasopressor requirement (AUC 0.700 [0.471–0.928]. None of the tested biomarkers proved useful for the prediction of mechanical ventilation within 48 h, with a combination of PCT and MR pro-ADM as best candidates only reaching an AUC of 0.553 (0.301–0.804). The combination of penKid, PCT and MR pro-ADM showed the best predictive performance for RRT (AUC 0.788 [0.358–1.000]).

Only MR pro-ADM showed potential as a predictor of 28-day mortality (AUC 0.686 [0.461–0.910], hospital mortality (AUC 0.699 [0.466–0.932]), and LOS on ICU; bio-ADM (in combination with pro-ADM) contributed to prediction of hospital LOS (**Table 4**).

## Discussion

This analysis aimed to assess the risk of AKI and to stratify patients in a cohort of ED all-comers with suspected organ dysfunction based on qSOFA, primarily included in the LifePOC study [26], which aimed at the early diagnosis of imminent

**Table 4. AUC of biomarkers to predict secondary endpoints (n = 440).**

| Secondary endpoints | Biomarker | AUC (95% CI) |
|---|---|---|
| **Endpoints within 48 h** | | |
| **Vasopressors/inotropes** | MR- proADM | 0.700 (0.471-0.928) |
| **Mechanical ventilation** | PCT, MR pro-ADM | 0.553 (0.302-0.804) |
| **RRT** | penKid, PCT, MR-proADM | 0.788 (0.358-1.000) |
| **Secondary endpoints within 28 days** | | |
| **28-day mortality** | MR pro-ADM | 0.686 (0.461-0.910) |
| **Hospital mortality** | MR pro-ADM | 0.699 (0.466-0.932) |
| **LOS ICU** | MR pro-ADM | 0.039* |
| **LOS hospital** | MR pro-ADM, bio-ADM | 0.028* |

**Abbreviations:** AUC: area under the curve; MR-proADM: midregional proadrenomedullin; PCT: procalcitonin; penKid: proenkephalin A119-159; bio-ADM: bioactive adrenomedullin; RRT: renal replacement therapy; LOS: length of stay

*: Coefficient of determination ($R^2$) of the multiple linear regression model (goodness of fit)

sepsis. We focussed on specific risk factors and on five biomarkers that may influence kidney function via different molecular pathways. Particular attention was paid to the novel biomarker penKid, for which increased levels have been shown to indicate impaired kidney function early [15,16]. Using multivariate analysis, we could identify pre-existing CKD, mechanical ventilation, increased levels of penKid at admission and the occurrence of sepsis within 48 hours as potential risk factors for the development of AKI within 48 hours. Our study focused on characterizing ED patients with various clinical pictures and an overall low disease severity (with a qSOFA score of 1 in 75.7% of the 440 patients available for analysis of the primary endpoint, AKI within 48 h, **Table 1**). In the ED context, a substantial proportion of AKI may reflect prerenal changes with early hypoperfusion, whereas excessive fluid administration can produce venous congestion. Restrictive fluid therapy, as could be reproduced in our data, may lower AKI risk because it prevents renal venous congestion and interstitial oedema, thereby preserving microvascular flow and oxygen delivery. This aligns with the current understanding that septic AKI arises primarily from microcirculatory and inflammatory mechanisms rather than sheer hypoperfusion.

The subsetting to patients with documented AKI status over 48 hours led to an obvious selection bias, which is evident in higher disease severity and particularly prominent regarding sepsis diagnosis. However, these results might also highlight sepsis as one of the most common causes of AKI and its pathogenesis which is distinct from purely ischemic or nephrotoxic injury often showing relatively little structural necrosis but significant functional impairment due to cellular and microcirculatory disturbances. Sepsis-induced AKI is driven by a combination of hemodynamic instability, microvascular dysfunction, dysregulated inflammation, and tubular metabolic changes.

Neutrophil Gelatinase-Associated Lipocalin (NGAL), Kidney Injury Molecule-1 (KIM-1), and cystatin C remain among the most validated and widely studied biomarkers for early AKI detection, with meta-analyses showing high diagnostic accuracy, especially for NGAL in both medical and surgical patients [33–35]. These markers are approved for clinical use in specific settings and have robust evidence supporting their predictive value for AKI occurrence and severity [33,36].

The biomarkers studied here complementarily reflect injury to the kidneys and the cardiovascular system, which are linked by different molecular pathways. To our knowledge, they have not been studied in this way before. We would like to emphasize that it was not our goal to identify their advantage over previously identified biomarkers, such as cystatin C, NGAL or KIM-1, for predicting AKI. The selection of these markers was based on different underlying pathologies leading to kidney dysfunction at different levels, e.g., endothelial dysfunction, modulation of the RAAS system and inflammation. These processes can all contribute to renal damage; markers linked to respective pathways might therefore be used as early markers of kidney dysfunction and provide pathophysiological information. From a pathophysiological point of view,

organ dysfunction can not only result from local damage, but also from systemic effects, e.g., via organ-organ interactions. Thus, AKI could trigger multi-organ failure and vice versa. We therefore not only included biomarkers reflecting the "true GFR" [16], but also markers regulating endothelial function [17,18]. Our results indicate that penKid and bio-ADM can predict AKI with a moderate diagnostic quality within 48 hours, independent of disease severity and pre-existing renal function. Prognostic performance of all biomarkers tested decreased with time to onset of AKI. In all constellations tested, penKid proved to be slightly superior to Scr and we thus conclude that penKid might indicate deterioration of renal function earlier than Scr, although differences in the current study were minor. Novel biomarkers for early prediction of deteriorating renal function might help to improve clinical management, as creatinine values only increase noticeably when the glomerular filtration rate is restricted by more than 50%. A "normal" creatinine value therefore does not rule out the onset of renal impairment. Moreover, Scr is influenced by multiple factors and is thus more suitable for monitoring the progression of renal dysfunction than for diagnosis. To further assess the differences between Scr and penKid, we compared two different eGFR equations: the formula $eGFR_{PENK-Crea}$, developed by Beunders and colleagues [25], and $eGFR_{CKD-EPI}$. We could confirm slight superiority of $eGFR_{PENK-Crea}$ over $eGFR_{CKD-EPI}$. However, this comparison must be interpreted with caution in the clinical context because eGFR estimation equations should not be used for transient state conditions.

The occurrence of AKI has been recognized as a very strong predictor of mortality in ICU patients [37]. Whereas penKid may provide additional information on AKI development, MR pro-ADM was shown to be a predictor of 28-day and hospital mortality in our study, consistent with the literature on severely ill patients with sepsis [38] and septic shock [39].

CKD is a well-known risk factor for AKI [40]. Patients with a pre-existing moderate to severe CKD are more likely to develop an acute-on-chronic renal failure, which is also reflected in our data. However, if the use of biomarkers can improve further preventive strategies to minimize the progression of CKD in patients known to be at high risk remains to be determined. A prediction of AKI seems to be most useful in patients with apparently "healthy kidneys" at admission as the patient's further treatment strategy may be significantly influenced. Measurement of biomarkers predictive of deteriorating kidney function upon admission to the ED may influence the decision of the treating physician, i.e., to hospitalize patients that seem less sick according to current diagnostic standards, thereby aiding an adequate management of patients at risk. Potential positive effects of such an approach must be evaluated, especially regarding major adverse kidney events. Whereas penKid, bio-ADM and pro-ADM displayed at least equal predictive performance compared to Scr in our analysis, there is still room for improvement in the establishment of clinically valid and applicable markers or marker combinations as early warning signs of imminent AKI. However, a combination of clinical information along with biomarkers could be used to identify patients at risk [41] and improve the diagnostic accuracy, process of care as well as the assistance of AKI management [42]. In particular, novel biomarkers might facilitate early AKI diagnosis and management in high-risk of complex clinical scenarios. However, their integration into routine practice requires further validation for cost-effectiveness and impact on patient outcomes. Nevertheless, clinical judgment will remain essential for interpreting biomarker results in the context of patient history, comorbidities, and exposures, as biomarker levels can be affected by non-renal factors such as inflammation or infection [43,44]. Further research is needed to define the optimal use of biomarkers in routine practice and to demonstrate their impact on long-term clinical outcomes.

## Strengths and limitations

An important strength of the presented study is its reflection of the real ED situation, including patients from all socioeconomic backgrounds and presenting with various clinical pictures and therefore reflecting the multifactorial causes of AKI. However, kidney endpoints were not the primary objective of the LifePOC study and were therefore only insufficiently documented, leading to the exclusion of a substantial part of patients primarily enrolled. As outlined above, the exclusion of these patients resulted in a selection bias towards patients with higher disease severity and higher incidence of sepsis than in the LifePOC cohort. We pinpointed this weakness by comparing the overall cohort and the subset available for the NephroPOC analysis. Likewise, further endpoints of interest were not recorded, such as major adverse kidney events,

e.g., persistent renal dysfunction or persistent renal-replacement therapy. For detailed assessment, serial measurements of all biomarkers and Scr should have been carried out. Using Scr-based definitions of AKI may not be free from significant confounders, such as muscle mass, diet, or the use of nephrotoxic agents. The differences between the eGFR are interesting, as no gold standard is available, a comparison to assess their performance should be interpreted with caution.

Similarly, pre-existing CKD could only be defined by the CCI, as an initial, targeted evaluation of kidney function was beyond the scope of the LifePOC study. As in any observational study, we could only determine associations but no causal relationships.

## Conclusion

Identifying high risk populations based on clinical risk factors and biomarkers can improve quality of care by allowing targeted monitoring and preventive strategies. PenKid, bio-ADM and pro-ADM are potential markers for surveillance of AKI development and MR pro-ADM a predictor of mortality. Prospectively, continued discovery and validation of renal biomarkers will be critical to their successful implementation for earlier and improved diagnosis and thus timely therapeutic intervention.

## Supporting information

**S1 Text. Supplemental digital content.**
(DOCX)

## Acknowledgments

The authors thank the staff of the Emergency Department of the Charité University Hospital in Berlin in both sites and of the University Hospital of Jena. Myrto Bolanaki and Peter Hajdu were responsible for LifePOC data acquisition in Berlin, whereas in Jena data acquisition was mainly done by Angelika Stacke, Caroline Neumann, Thomas Fricke and Margarita Suitchmezian. Special thanks to the study nurses Sandra Signert and Carine Werk-Wenzel as well as to Cora Richert and Kay Stötzer for their technical support and to all students that contributed to the recruitment and data acquisition.

## Author contributions

**Conceptualization:** Caroline Neumann.

**Formal analysis:** Margit Leitner, Thomas Lehmann.

**Methodology:** Caroline Neumann, Margit Leitner, Thomas Lehmann.

**Project administration:** Caroline Neumann.

**Supervision:** Michael Joannidis.

**Validation:** Caroline Neumann, Margit Leitner.

**Visualization:** Margit Leitner.

**Writing – original draft:** Caroline Neumann.

**Writing – review & editing:** Caroline Neumann, Margit Leitner, Michael Kiehntopf, Michael Joannidis, Myrto Bolanaki, Anna Slagman, Martin Möckel, Michael Bauer, Johannes Winning.

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
