## [Decision Letter · Decision Letter 0]

3 Nov 2025

PONE-D-25-48342NephroPOC - Risk assessment and prediction of acute kidney injury in emergency patients with suspected organ dysfunction: secondary analysis from the prospective observational LifePOC studyPLOS ONE

Dear Dr. Neumann,

Thank you for submitting your manuscript to PLOS ONE. After careful consideration, we feel that it has merit but does not fully meet PLOS ONE’s publication criteria as it currently stands. Therefore, we invite you to submit a revised version of the manuscript that addresses the points raised during the review process.

This is an interesting novel biomarkers for acute kidney injury with well design study. There are only minor comments from the reviewers. Please carefully response to the reviewers' comments and suggestions.

We look forward to receiving your revised manuscript.

Kind regards,

Vipa Thanachartwet, M.D.

Academic Editor

PLOS ONE

Journal Requirements:

3. Please be informed that funding information should not appear in the Acknowledgments section or other areas of your manuscript. We will only publish funding information present in the Funding Statement section of the online submission form. Please remove any funding-related text from the manuscript.

Reviewers' comments:

Reviewer's Responses to Questions

**Comments to the Author**

1. Is the manuscript technically sound, and do the data support the conclusions?

Reviewer #1: Yes

Reviewer #2: Yes

2. Has the statistical analysis been performed appropriately and rigorously? 

Reviewer #1: I Don't Know

Reviewer #2: Yes

3. Have the authors made all data underlying the findings in their manuscript fully available?

Reviewer #1: Yes

Reviewer #2: Yes

4. Is the manuscript presented in an intelligible fashion and written in standard English?

Reviewer #1: Yes

Reviewer #2: Yes

5. Review Comments to the Author

Reviewer #1: Thank you for submitting this article. It is interesting to read about this specific biomarker. I think the discussion could include more about the benefits of the use of the biomarker over clinician knowledge.

Reviewer #2: PLOS ONE

October 1, 2025

Manuscript ID: PONE-D-25-48342

Manuscript title: NephroPOC - Risk assessment and prediction of acute kidney injury in emergency patients with suspected organ dysfunction: secondary analysis from the prospective observational LifePOC study

Dear Prof. Thanachartwet,

This study was conducted as a well design study with the aim to assess markers and risk factors for imminent acute kidney injury (AKI) in emergency patients. The results from this study could be used as a novel biomarkers for predicting acute kidney injury in future.

However, there are some comments for this manuscript to address as follows:

1. Currently, there are several biomarkers for AKI. Are there any evidence to support that biomarkers including penKid, ADM, DPP-3 and PCT have the advantage over the previous biomarkers such as Cystatin C, NGAL, KIM-1 or etc. for predicting AKI?

2. Regarding Table S5: Prediction of AKI within 48h using admission biomarkers and GFR estimates: Net reclassification improvement based on GFR categories, this table compared eGFR by CKD-EPI and PENK-based equation according to REF #25. This PENK-based equation arise from the international multicenter study of 1,354 stable and critically ill patients with stable kidney function. In this study, GFR was measured (mGFR) through iohexol or iothalamate clearance which is the gold standard method for measuring GFR. It is wonder how PENK-based equation can be used to estimate GFR in AKI patients. Additionally, how CKD-EPI can estimate true GFR in AKI patients as CKD-EPI is not gold standard for measuring GFR? Thus, eGFR categories by PENK-Crea and CKD-EPI in the Tale S5 should be analysed by the test of agreement.

3. P-value throughout the manuscript should be round to the 3 digits after the decimal point.

6. PLOS authors have the option to publish the peer review history of their article (what does this mean? ). If published, this will include your full peer review and any attached files.

**Do you want your identity to be public for this peer review?** For information about this choice, including consent withdrawal, please see our Privacy Policy .

Reviewer #1: No

Reviewer #2: **Yes:**  Assoc. Prof. Varunee Desakorn

---

## [Author Response · Author response to Decision Letter 1]

12 Nov 2025

Dear Prof. Thanachartwet,

Dear Reviewers,

Thank you very much for your evaluation, with clear and constructive criticism, which was very helpful to improve the quality of our manuscript.

We hope to have addressed all issues adequately and that the revised manuscripts is now suitable for the publication in “PlosOne”.

Yours sincerely,

Caroline Neumann.

Response to Editor

Response: Thank you for pointing this shortcoming out. We adapted the manuscript according to the template.

3. Please be informed that funding information should not appear in the Acknowledgments section or other areas of your manuscript. We will only publish funding information present in the Funding Statement section of the online submission form. Please remove any funding-related text from the manuscript.

Response to 2. and 3.: We could not detect the said discrepancies. The disclosure statements include any funding that could be perceived to cause a conflict of interest, whereas the funding information only refers to the grant that supported work on the manuscripts topic. However, as funding information should not be included in the manuscript (see 3.), we deleted the respective sections and provide the relevant information in the online submission form only.

Response: We removed the ethics statement in the Declarations section. All pertaining information is given in the Methods section.

Response: We included the captions and renumbered all supplementary materials to conform to the guidelines for supporting information files.

Response: Thank you for this information.

Response: We reviewed our reference list and confirm that it is complete and correct. No retracted papers have been cited. However, we added some further references during the revision in the paragraphs added to the manuscript.

Response to reviewers

Reviewer #1: Thank you for submitting this article. It is interesting to read about this specific biomarker. I think the discussion could include more about the benefits of the use of the biomarker over clinician knowledge.

Response: Thank you for mentioning this important issue. We have now adjusted the discussion accordingly and addressed this relevant topic. Compared to clinical knowledge and risk scores, which rely on clinical variables and delayed functional markers, novel biomarkers may provide earlier risk stratification, improve prognostic enrichment, and have potential for guiding timely interventions. However, their integration into routine practice requires further validation for cost-effectiveness and impact on patient outcomes. Nevertheless, clinical judgment is essential for interpreting biomarker results in the context of patient history, comorbidities, and exposures, as biomarker levels can be affected by non-renal factors such as inflammation or infection (PMID: 40293565, PMID: 22650449). Further research is needed to define the optimal use of biomarkers in routine practice and to demonstrate their impact on long-term clinical outcomes.

Reviewer #2

October 1, 2025

Manuscript ID: PONE-D-25-48342

Manuscript title: NephroPOC - Risk assessment and prediction of acute kidney injury in emergency patients with suspected organ dysfunction: secondary analysis from the prospective observational LifePOC study

Dear Prof. Thanachartwet,

This study was conducted as a well design study with the aim to assess markers and risk factors for imminent acute kidney injury (AKI) in emergency patients. The results from this study could be used as a novel biomarkers for predicting acute kidney injury in future.

However, there are some comments for this manuscript to address as follows:

1. Currently, there are several biomarkers for AKI. Are there any evidence to support that biomarkers including penKid, ADM, DPP-3 and PCT have the advantage over the previous biomarkers such as Cystatin C, NGAL, KIM-1 or etc. for predicting AKI?

Response: Thank you very much for raising this important concern. Neutrophil Gelatinase-Associated Lipocalin (NGAL), Kidney Injury Molecule-1 (KIM-1), and cystatin C remain among the most validated and widely studied biomarkers for early AKI detection, with meta-analyses showing high diagnostic accuracy, especially for NGAL in both medical and surgical patients (PMID: 36371256, PMID: 26860999, PMID: 33556265). These markers are approved for clinical use in specific settings and have robust evidence supporting their predictive value for AKI occurrence and severity (PMID: 36371256, PMID: 39826969).

The biomarkers studied here complementarily reflect injury to the kidneys and the cardiovascular system, which are linked by different molecular pathways. To our knowledge, they have not been studied in this way before. We would like to emphasize that it was not our goal to identify their advantage over previous biomarkers such as cystatin C, NGAL, KIM-1, etc. for predicting AKI. Rather, the rationale behind the selection of these markers were different underlying pathologies leading to kidney dysfunction at different levels as endothelial function, the modulation of the RAAS system and inflammation all contribute to renal damage and might therefore be used as early kidney markers as well. Thank you again for your clarification, we highlight this point now specifically in the revised discussion.

2. Regarding Table S5: Prediction of AKI within 48h using admission biomarkers and GFR estimates: Net reclassification improvement based on GFR categories, this table compared eGFR by CKD-EPI and PENK-based equation according to REF #25. This PENK-based equation arise from the international multicenter study of 1,354 stable and critically ill patients with stable kidney function. In this study, GFR was measured (mGFR) through iohexol or iothalamate clearance which is the gold standard method for measuring GFR. It is wonder how PENK-based equation can be used to estimate GFR in AKI patients. Additionally, how CKD-EPI can estimate true GFR in AKI patients as CKD-EPI is not gold standard for measuring GFR? Thus, eGFR categories by PENK-Crea and CKD-EPI in the Tale S5 should be analysed by the test of agreement.

Response: Thank you for this important hint. We agree that measurement of GFR by iohexol or iothalamate clearance remains the gold standard for assessing kidney function. Regarding your suggestion to analyse the agreement between eGFR categories by PENK-Crea and CKD-EPI using a test of agreement, we performed Cohen’s Kappa analysis. The resulting Kappa coefficients of 0.672 (95% confidence interval: 0.534 to 0.786) in patients with AKI within 48h and 0.626 (95% confidence interval: 0.559 to 0.686) in patients without AKI within 48h indicates substantial agreement between the two classification methods according to commonly accepted interpretation criteria (Landis JR, Koch GG. Biometrics. 1977 Mar;33(1):159-74.). We already addressed this point in the discussion (“However, this comparison must be interpreted with caution in the clinical context because eGFR estimation equations should not be used for transient state conditions.”) and now added the above-mentioned analysis to the manuscript.

3. P-value throughout the manuscript should be round to the 3 digits after the decimal point.

Response: Thank you very much for pointing out the nuances. We have adjusted the p-values round to the 3 digits after the decimal point throughout the manuscript and the supplement.

---

## [Decision Letter · Decision Letter 1]

11 Dec 2025

NephroPOC - Risk assessment and prediction of acute kidney injury in emergency patients with suspected organ dysfunction: secondary analysis from the prospective observational LifePOC study

PONE-D-25-48342R1

Dear Dr. Neumann,

We’re pleased to inform you that your manuscript has been judged scientifically suitable for publication and will be formally accepted for publication once it meets all outstanding technical requirements.

Kind regards,

Vipa Thanachartwet, M.D.

Academic Editor

PLOS One

Additional Editor Comments (optional):

All comments have been addressed.

Reviewers' comments:

Reviewer's Responses to Questions

**Comments to the Author**

1. If the authors have adequately addressed your comments raised in a previous round of review and you feel that this manuscript is now acceptable for publication, you may indicate that here to bypass the “Comments to the Author” section, enter your conflict of interest statement in the “Confidential to Editor” section, and submit your "Accept" recommendation.

Reviewer #1: All comments have been addressed

Reviewer #2: All comments have been addressed

2. Is the manuscript technically sound, and do the data support the conclusions?

Reviewer #1: Yes

Reviewer #2: Yes

3. Has the statistical analysis been performed appropriately and rigorously? 

Reviewer #1: I Don't Know

Reviewer #2: Yes

4. Have the authors made all data underlying the findings in their manuscript fully available?

Reviewer #1: Yes

Reviewer #2: Yes

5. Is the manuscript presented in an intelligible fashion and written in standard English?

Reviewer #1: Yes

Reviewer #2: Yes

6. Review Comments to the Author

Reviewer #1: Thank you for making the suggested changes. This is an interesting article and I look forward to seeing it in print

Reviewer #2: (No Response)

7. PLOS authors have the option to publish the peer review history of their article (what does this mean? ). If published, this will include your full peer review and any attached files.

**Do you want your identity to be public for this peer review?** For information about this choice, including consent withdrawal, please see our Privacy Policy .

Reviewer #1: No

Reviewer #2: **Yes:**  Assoc Prof. Varunee Desakorn

---

## [Editor Report · Acceptance letter]

PONE-D-25-48342R1

PLOS One

Dear Dr. Neumann,

I'm pleased to inform you that your manuscript has been deemed suitable for publication in PLOS One. Congratulations! Your manuscript is now being handed over to our production team.

Kind regards,

on behalf of

Professor Vipa Thanachartwet

Academic Editor

PLOS One